# Synthesis of Organic Semiconductor Nanoparticles with Different Conformations Using the Nanoprecipitation Method

**DOI:** 10.3390/polym14245336

**Published:** 2022-12-07

**Authors:** Nathalia A. Yoshioka, Thales A. Faraco, Hernane S. Barud, Sidney J. L. Ribeiro, Marco Cremona, Benjamin Fragneaud, Indhira O. Maciel, Welber G. Quirino, Cristiano Legnani

**Affiliations:** 1Grupo de Nanociência e Nanotecnologia (NANO), Departamento de Física, Universidade Federal de Juiz de Fora (UFJF), Juiz de Fora 36036-330, MG, Brazil; 2Laboratório de Biopolímeros e Biomateriais (BIOPOLMAT), Departamento de Química, Universidade de Araraquara (UNIARA), Araraquara 14801-340, SP, Brazil; 3Laboratório de Materiais Fotônicos, Instituto de Química, Universidade Estadual Paulista (UNESP), Araraquara 14800-060, SP, Brazil; 4Laboratório de Optoeletrônica Molecular (LOEM), Departamento de Física, Pontifícia Universidade Católica do Rio de Janeiro (PUC-Rio), Rio de Janeiro 22453-970, RJ, Brazil

**Keywords:** organic semiconductor nanoparticles, P3HT, PC_71_BM, nanoprecipitation method, aqueous suspensions

## Abstract

In recent years, nanoparticulate materials have aroused interest in the field of organic electronics due to their high versatility which increases the efficiency of devices. In this work, four different stable conformations based on the organic semiconductors P3HT and PC_71_BM were synthesized using the nanoprecipitation method, including blend and core-shell nanoparticles. All nanoparticles were obtained free of surfactants and in aqueous suspensions following the line of ecologically correct routes. The structural and optoelectronic properties of the nanoparticles were investigated by dynamic light scattering (DLS), transmission electron microscopy (TEM), scanning electron microscopy (SEM), UV-visible absorption spectroscopy and UV-visible photoluminescence (PL). Even in aqueous media, the blend and core-shell nanoparticles exhibited a greater light absorption capacity, and these conformations proved to be effective in the process of dissociation of excitons that occurs at the P3HT donor/PC_71_BM acceptor interface. With all these characteristics and allied to the fact that the nanoparticles are surfactant-free aqueous suspensions, this work paves the way for the use of these colloids as a photoactive layer of organic photovoltaic devices that interface with biological systems.

## 1. Introduction

For many years, inorganic photovoltaic devices based on crystalline silicon semiconductor films [1] have dominated the scenario of scientific research and industrial production. In addition to exhibiting high efficiency in converting solar energy into electrical energy, they are in accordance with the demands of production of sustainable energy, contributing to the reduction of environmental impacts that have been widely discussed by civil, scientific, industrial and governmental societies.

Over the last few decades, organic photovoltaic devices (OPVs) [2] based on organic semiconductor materials (conjugated polymers and small molecules) have stood out due to the superior characteristics they exhibit in relation to conventional ones, such as mechanical flexibility, semi-transparency, lightness, low-cost and easy processing [3,4,5,6]. These characteristics increase the applicability of OPVs in various fields of science.

One of the most investigated OPVs consists of a photoactive layer based on a bulk heterojunction (BHJ) of a conjugated polymer, poly(3-hexylthiophene) (P3HT), and a fullerene derivative, (6,6)-Phenyl C_61_ butyric acid methyl ester (PC_61_BM) [7,8]. P3HT is an electron donor and PC_61_BM acts as an electron acceptor. Excitons (electron-hole pairs) are produced in the P3HT polymer chains and diffuse to the PC_71_BM interface. This BHJ allows effective dissociation of excitons at the donor/acceptor interface before the natural recombination process occurs, improving the efficiency of P3HT-only OPVs.

These OPVs have low average power conversion efficiency (PCE) of up to ~5% [9,10]. However, the use of various organic and inorganic nanostructured materials inserted into the P3HT:PC_61_BM photoactive layer has been reported in the literature [11,12,13,14,15,16,17] to improve the efficiency of these OPV devices. Furthermore, P3HT and PC_61_BM nanoparticles have been used with the main objective of increasing the interfacial areas of exciton dissociation in the photoactive layer to improve the PCEs of OPVs [18,19,20,21].

As organic materials are basically formed by carbon and hydrogen atoms, a structural affinity with biological systems is expected [22], allowing important biomedical applications through the construction of devices based on these materials.

Recent studies have shown that P3HT can act directly at the cellular interface for neuronal photovoltaic stimulation through light impulses [23,24,25,26]. Degeneration of retinal photoreceptors is one of the most frequent causes of progressive visual field loss in humans, usually caused by diseases such as retinitis pigmentosa, Stargardt disease and age-related macular degeneration [27]. There is no known cure for degenerative retinal diseases, and existing therapies only act by slowing down their progression. In this sense, the development of retinal implantable devices based on organic semiconductors represents an important path towards the possibility of restoring visual acuity to patients with retinal degeneration.

The method used to obtain nanoparticles is an important requirement that must be considered in order to allow the formation of stable colloidal suspensions with specific characteristics for the desired applications. The nanoprecipitation method, also called solvent displacement [28,29], is an easy, reproducible and low-cost route that is capable of producing stable colloids in water without the need for surfactants or other additives. These suspension-related characteristics make the nanoprecipitation more appropriate to obtain colloids for biological applications than its main competitor, the miniemulsion method [30,31,32].

The nanoprecipitation method consists of an organic phase (composed of polymers dissolved in a good solvent) and a non-organic phase (composed of a poor solvent). The colloidal suspension of nanoparticles is formed when the organic phase is added to the aqueous phase under agitation and due to the solvent-non-solvent surface tension generated by the difference in polarity between these solvents together with their miscibility. Experimental parameters must be strictly controlled since they directly influence the formation of NPs [33]. 

In this work, we synthesized nanoparticles (NPs) using the nanoprecipitation method to obtain stable colloidal suspensions in water and free of surfactants. We were able to produce four different conformations: P3HT nanoparticles, PC_71_BM nanoparticles, P3HT:PC_71_BM blend nanoparticles and P3HT©PC_71_BM core-shell nanoparticles (where P3HT is the core and PC_71_BM is the shell). The structural and optoelectronic properties of the four conformations of NPs were investigated. 

## 2. Materials and Methods

### 2.1. Materials

Poly(3-hexylthiophene-2,5-diyl) (P3HT) (regioregularity > 93%, Mw > 45,000 g/mol) and (6,6)-Phenyl C_71_ butyric acid methyl ester (PC_71_BM) (>99%, Mw 1030.93 g/mol) were both obtained from Lumtec Co. (Taipei, Taiwan), while the solvents tetrahydrofuran (THF) and dimethyl sulfoxide (DMSO) were purchased from Vetec, Sigma-Aldrich (São Paulo, Brazil).

### 2.2. Synthesis of the Organic Nanoparticles

We prepared four different conformations of NPs using the nanoprecipitation method and obtained very stable colloidal suspensions. Figure 1 shows a scheme of those NPs: (a) P3HT; (b) PC_71_BM nanoparticles; (c) NPs blend of P3HT and PC_71_BM (denoted by P3HT:PC_71_BM); and (d) core-shell NPs with a P3HT core and a PC_71_BM shell (denoted by P3HT©PC_71_BM).

It should be noted that we are interested in a PCBM-based shell and not a P3HT-based shell since the PCBM is an electron acceptor.

For P3HT and PC_71_BM NPs, 0.5 mg/mL stock solutions of P3HT and PC_71_BM in THF were heated for 1 h and 30 min at 50 °C. To obtain the colloidal suspension of P3HT NPs, this solution was mixed with deionized water at a volume ratio of 1:9 (*v*/*v*) while stirring the mixture at 500 rpm for 15 s at a temperature of 30 °C. The same procedure was performed for the synthesis of PC_71_BM NPs. For colloids of P3HT:PC_71_BM blend and P3HT©PC_71_BM core-shell nanoparticles, the stock solution was prepared by dissolving 5.0 mg of a mixture of P3HT and PC_71_BM in a weight ratio of 1:1 (*w*/*w*) in 10.0 mL of THF. P3HT:PC_71_BM blend NPs were synthesized by mixing the P3HT/PC_71_BM stock solution with deionized water (1:9 *v*/*v*) (Figure 2a). The colloidal suspension of P3HT©PC_71_BM core-shell NPs was produced as follows: The P3HT/PC_71_BM stock solution in THF was dropped into DMSO (1:9 *v*/*v*) (DMSO slightly dissolves PC_71_BM but cannot dissolve P3HT) forming a colloid of P3HT NPs in DMSO. Finally, this suspension was then mixed with deionized water while stirring at a volume ratio of 1:9 (*v*/*v*) creating a PC_71_BM shell covering the P3HT nanoparticles. In all steps, the solution was stirred at 500 rpm for 15 s at 30 °C (Figure 2b). 

We have observed that the volume ratio of 1:9 (*v*/*v*) is better for obtaining stable colloids with smaller nanoparticle diameters. When we increased the volume of the solvent phase, we observed the formation of larger structures or even clusters of nanoparticles. In addition, we used P3HT with a molecular weight of >45,000 g/mol. It is possible that a lower molecular weight would produce smaller nanoparticles, but it would imply on shorter conjugated chains which would affect the absorption range of the nanoparticles and consequently their photoconductivity.

After the synthesis of all colloidal suspensions, the colloids were rota-evaporated at 60 °C for 20 min in order to evaporate the remaining THF. DMSO has a higher boiling point (189 °C), and it is not evaporated by the rota-evaporating process. However, the biological applications of core-shell NPs are not affected by the presence of DMSO in colloidal suspensions since DMSO is widely used in the preservation of several types of cells, proving to be biocompatible [34].

The NPs were deposited by spray coating onto indium tin oxide (ITO) coated glass substrates at 150 °C for 1 s using an airbrush with a 0.3 mm nozzle (Steula, BC 62).

### 2.3. Characterization Methods

The particle sizes were determined by a Malvern Zetasizer Nano ZS90 analyzer using the dynamic light scattering (DLS) technique. Transmission electron microscopy (TEM) images of nanoparticles were obtained using a Philips CM200 transmission electron microscope. Scanning electron microscopy (SEM) images of nanoparticles on ITO substrates were obtained using a JEOL JSM-7500F scanning electron microscope. The absorption and emission spectra were obtained by a Shimadzu UV-1800 spectrophotometer and a PTI QuantaMaster 40 spectrofluorometer, respectively.

## 3. Results and Discussion

### 3.1. Structural Analysis

The average diameter of the NPs synthesized in this work were analyzed by DLS and the results are shown in Table 1. The polydispersity index (PDI) is an important statistic parameter for observing the breadth or dispersion of NP size distribution. The PDI is calculated by dividing the square of the standard deviation of the diameter by the square of the mean diameter. 

The numerical value of the PDI ranges from 0 to 1, and values close to zero indicate particles with more homogeneous size distribution, i.e., highly monodisperse. Values close to 1 indicate polydispersity with populations of various particle sizes (ISO 22412:2017) [35]. Values below 0.4 are commonly considered acceptable for monodisperse polymer-based nanoparticles [36] and as can be seen in Table 1, all of our colloidal suspensions had PDI values below 0.3. This slight polydispersity can be an advantage when you want to deposit NPs in the form of thin films as the smaller nanoparticles can occupy the regions between the larger ones, effectively covering the substrate.

The TEM images of the synthesized NPs are shown in Figure 3, where it is noticeable that the four types of synthesized NPs have mostly spherical morphology. These images show mostly individual NPs, attesting to the effectiveness of the suspensions, which is important to obtain homogeneous NP-based films. The mean diameters of the NPs confirm the results obtained by DLS measurements.

As shown in Figure 3d, the darker core and the lighter shell of the P3HT©PC_71_BM NPs suggest the formation of a core-shell structure. As the P3HT is thicker than the PC_71_BM and has heavier sulfur atoms in its structure, the regions where this polymer is present provide a greater electron scattering and appear darker in the TEM images [37,38]. This suggests that the core of the nanoparticles is composed of P3HT and the shell of PC_71_BM.

ITO is a n-type semiconductor thin film and widely used for the construction of organic photovoltaic devices (OPVs) [39] due to its high visible transmittance and low electrical resistivity (10^−4^ Ω·cm) [40]. Therefore, it is important to observe how the NPs are going to be assembled onto this substrate since this is one of the standard architectures for OPVs. 

Figure 4 shows SEM images of the P3HT©PC_71_BM core-shell NPs deposited on ITO substrates by spray coating [41]. The darker regions and their contours are where the NPs were deposited. It is important to notice that this is not a usual method to obtain films for optoelectronics. Normally NPs are deposited by spin coating. However, this method usually requires the use of a volatile solvent; however, our suspensions are aqueous which makes it difficult to obtain good coverage of the substrate [42].

Even though one observes only a partial coverage of the substrate, it is important to state that this method is efficient for depositing films that preserve the NP shape of the materials. A better coverage can be achieved by repeating the spray coating, as can be observed by comparing Figure 4a,b, where P3HT©PC_71_BM core-shell NPs were deposited by one and three steps, respectively. Furthermore, the colloidal suspensions were very diluted, meaning that more concentrated suspensions might also improve coverage.

### 3.2. Optical Measurements

UV-visible spectra of the stock solutions in THF and the colloids in aqueous suspensions are shown in Figure 5. A blank measurement was performed for control and took into account the proportions of water and DMSO used during the synthesis.

Compared with the optical absorption spectra of the stock solutions, the colloidal suspensions exhibited a redshift of UV-Vis absorption maxima (bathochromic shifts) accompanied by a broadening of the bands. The largest shifts to longer wavelengths were observed for colloids containing P3HT.

When P3HT is dissolved in a good solvent (stock solution), the polymer chains are fully extended in random-coil conformation. As NPs are formed, the polymer chains approach each other and tend to organize into a more planar backbone conformation, which leads to an increase in the length of conjugation and delocalized electrons. With this lower energy configuration, the absorption spectra are redshifted. Two shoulders appear in the spectra of the colloidal suspensions and are associated with strong π-π* intrachain (J-aggregates) and interchain (H-aggregates) interactions between the thiophene rings of P3HT [43,44]. The 0-0 transitions, related to J-aggregates, occur at 600 nm, while the 0-1 transitions occur at 550 nm and indicate interchain transitions in H-aggregates [44,45,46,47]

Furthermore, deviations in the planarity of the chains may be also related to the broadening of the optical absorption spectrum [48]. 

Figure 6a,b show the photoluminescence (PL) spectra of the P3HT, P3HT:PC_71_BM blend and P3HT©PC_71_BM core-shell NPs excited with radiation at 510 nm and 560 nm. The wavelength of 510 nm was chosen because it is near to the maximum absorption of the NPs that contain P3HT, while 560 nm is a wavelength where the light is still being absorbed, but with a lower intensity, and it is related to the most energetic interchain transitions between P3HT molecules.

In both graphs, it is possible to notice the quenching of PL when PC_71_BM is added to the nanoparticles. As already mentioned, P3HT and PC_71_BM are organic semiconductors, such that P3HT is an electron donor and PC_71_BM is an electron acceptor. By absorbing light, P3HT generates electron-hole pairs (excitons), and the PL process is related to the radiative recombination of those pairs. With the inclusion of PC_71_BM, excitons formed in the polymeric chains of P3HT diffuse to the donor/acceptor interface, as depicted in Figure 7a. Electrons are then transferred from the LUMO of the generating molecule to the LUMO of the acceptor molecule, while the holes remain in the HOMO of the donor material due to the potential barrier, as shown in Figure 7b. As the electron transfer process between different materials occurs much faster (in the order of femtoseconds) than the recombination process within the donor material, a drastic decrease in the photoemission process is observed [49,50] and explains the PL quenching.

## 4. Conclusions

We synthesized different stable nanoparticle conformations based on a P3HT conjugated polymer and a PC_71_BM fullerene derivative using the nanoprecipitation technique which proved to be an easy, reproducible, eco-friendly and low-cost method to obtain surfactant-free aqueous suspensions. Through the characterizations, it was possible to observe the dispersion of the size distribution of the nanoparticles which proved to have more monodisperse than polydisperse characteristics. The NPs were basically spherical and the core-shell structure with a P3HT core and a PC_71_BM shell could be easily observed through the TEM contrast differences. The absorption spectra of the colloidal suspensions showed a redshift and a broadening of the band in relation to the spectra of the stock solutions in THF, indicating an aggregation state of P3HT from amorphous to a more ordered configuration. The extinction of the emission band observed in the photoluminescence (PL) spectra evidenced the dissociation of excitons at the P3HT/PC_71_BM interface for blend and core-shell nanoparticles. Equivalently, these conformations of NPs allow an effective transfer of electrons from the donor material to the acceptor even when the NPs are in an aqueous suspension. These characteristics suggest that these nanoparticles can be successfully used as a photoactive layer in organic photovoltaic systems, especially regarding biomedical applications, such as in work that seeks to develop P3HT-based retinal implantable devices to restore visual acuity in people with retinal degeneration.

## Figures and Tables

**Figure 1 polymers-14-05336-f001:**
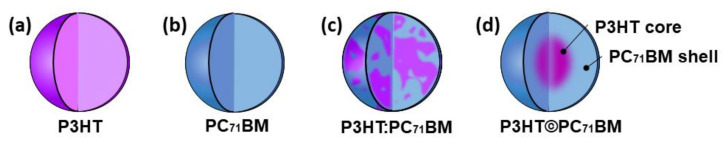
Schematic representation of: (**a**) P3HT, (**b**) PC_71_BM, (**c**) P3HT:PC_71_BM blend and (**d**) P3HT© PC_71_BM core-shell NPs.

**Figure 2 polymers-14-05336-f002:**
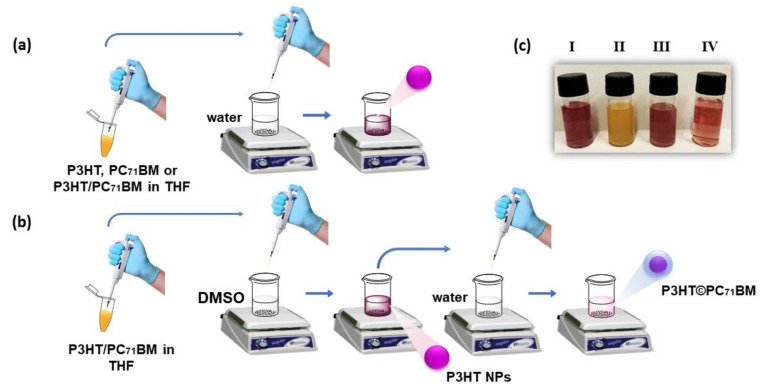
Schematic diagram of the steps involved in the nanoprecipitation method for the formation of (**a**) P3HT, PC_71_BM and P3HT:PC_71_BM blend NPs; and (**b**) P3HT©PC_71_BM core-shell NPs. (**c**) Pictures of (I) P3HT, (II) PC_71_BM, (III) P3HT:PC_71_BM blend and (IV) P3HT©PC_71_BM core-shell colloidal suspensions.

**Figure 3 polymers-14-05336-f003:**
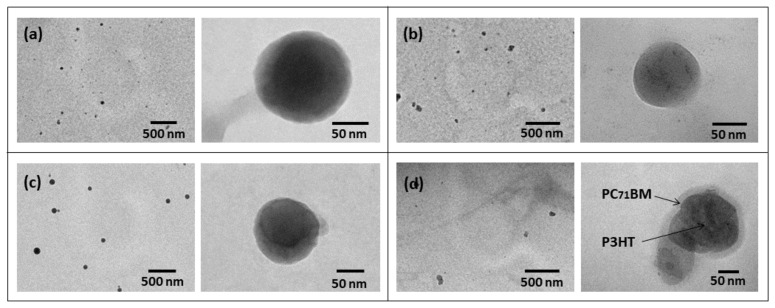
TEM images of (**a**) P3HT, (**b**) PC_71_BM, (**c**) P3HT:PC_71_BM blend and (**d**) P3HT©PC_71_BM core-shell NPs.

**Figure 4 polymers-14-05336-f004:**
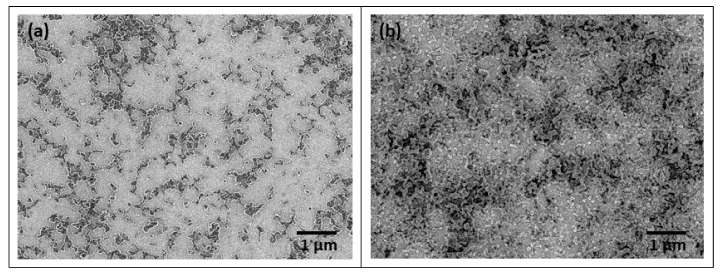
SEM images of P3HT©PC_71_BM core-shell NPs deposited on ITO substrate by (**a**) one and (**b**) three steps.

**Figure 5 polymers-14-05336-f005:**
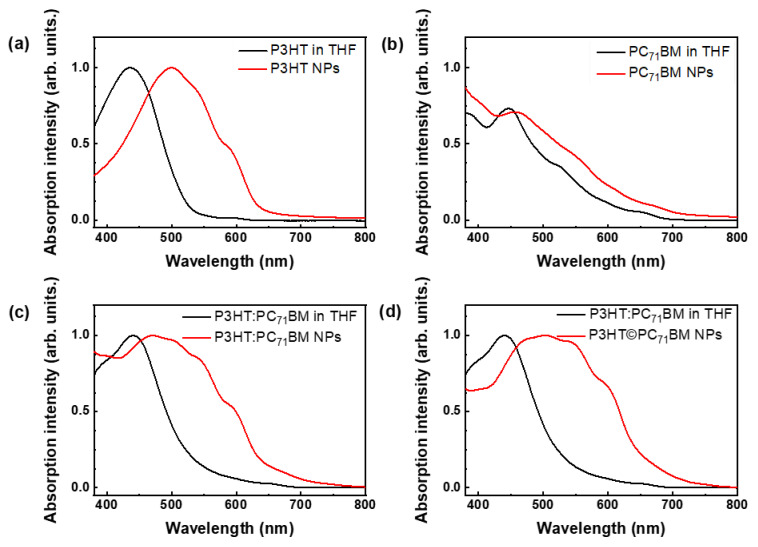
Absorption spectra of colloidal suspensions in water of (**a**) P3HT, (**b**) PC_71_BM, (**c**) P3HT:PC_71_BM blend and (**d**) P3HT©PC_71_BM core-shell NPs in red and a comparison to their stock solutions in THF in black.

**Figure 6 polymers-14-05336-f006:**
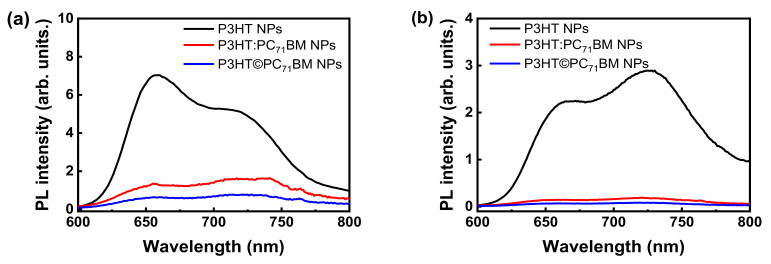
Photoluminescence spectra of aqueous suspensions of P3HT, P3HT:PC_71_BM blend and P3HT©PC_71_BM core-shell NPs excited with (**a**) 510 nm and (**b**) 560 nm radiation wavelengths.

**Figure 7 polymers-14-05336-f007:**
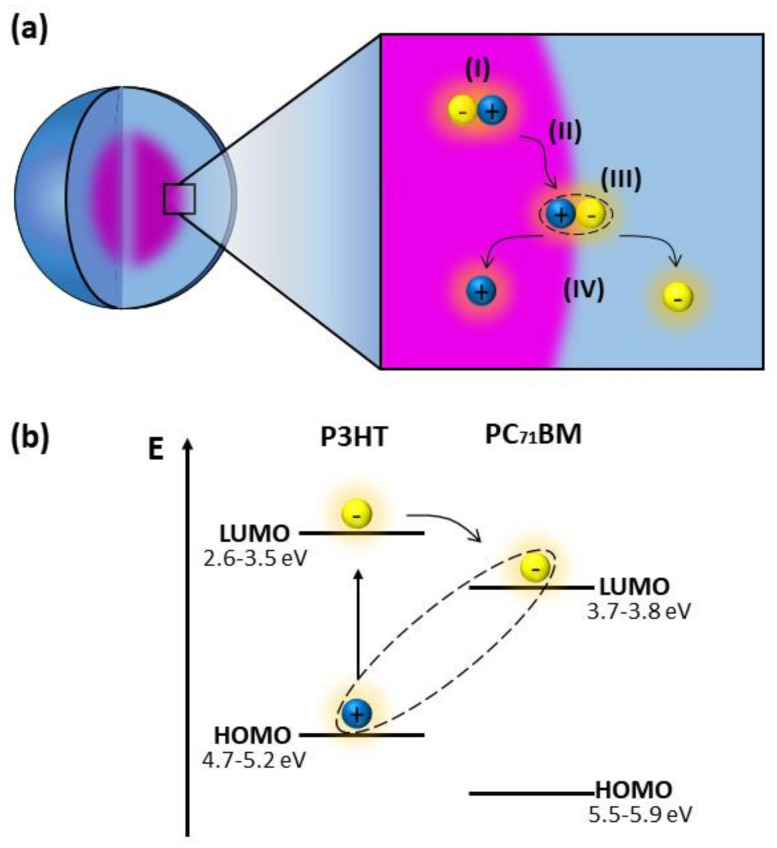
(**a**) Schematic representation of the processes of generation (I), diffusion (II), dissociation of exciton at the donor/acceptor interface (III) and charge separation (IV)—which occurs when an electric field is applied [51]. (**b**) Energy levels (HOMO and LUMO) showing the exciton dissociation at the P3HT/PC_71_BM interface.

**Table 1 polymers-14-05336-t001:** Average diameter and their respective PDI for different conformations of NPs analyzed via DLS.

Nanoparticles	Average Diameter (nm)	PDI
P3HT	(1.4 ± 0.6) × 10^2^	0.15
PC_71_BM	(1.3 ± 0.8) × 10^2^	0.23
P3HT:PC_71_BM	(1.4 ± 0.5) × 10^2^	0.09
P3HT©PC_71_BM	(2.0 ± 1.0) × 10^2^	0.26

## Data Availability

The data presented in this study are available on request from the corresponding author.

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
