# Peer review of "Synthesis of Organic Semiconductor Nanoparticles with Different Conformations Using the Nanoprecipitation Method"

_polymers, 2022, doi:10.3390/polym14245336_

Round 1
Reviewer 1 Report
In this manuscript, the authors synthesized different P3HT and PCBM nanoparticles and studied their morphological and optical properties. I suggest the acceptance of the manuscript after the following revision:
1. What's the molecular weight of P3HT? Will the molecular weight of P3HT affect the synthesized structure?
2. The author used 1:9 (v/v) for the synthesis, why this volume ratio was chosen? what's the morphology of the structures synthesized with other volume ratio?
3. For the P3HT/PCBM core/shell structure, what is the interaction between P3HT/PCBM that is holding them together? If you add P3HT later to the PCBM NP, will you form PCBM/P3HT core/shell structure?
4. I suggest the authors to conduct FTIR measurement to understand the interaction between P3HT and PCBM
5. This manuscript covers some important characterizations such as TEM, SEM, Absorption spectra, which is good. However, I think it's also important to show the readers that those nanoparticles can be applied in some devices. Can the authors also include the data of the application of the synthesized nanoparticles? for example in OPV or in biomedical applications as stated in the manuscript
6. It seems like the P3HT/PCBM nanostructures synthesized using the current method can not be spin-coated, and the aqueous NP suspension is difficult to form good coverage of the substrate, then can the authors give some suggestions about how to apply the synthesized particles in the fabrication of OPV device?
7. some literatures regarding the synthesis and application of P3HT/PCBM nanostructures can be cited: "Bottom-up approaches for precisely nanostructuring hybrid organic/inorganic multi-component composites for organic photovoltaics." MRS Advances, 5(40-41), pp.2055-2065.; "Facile synthesis of water-dispersible poly (3-hexylthiophene) nanoparticles with high yield and excellent colloidal stability." Iscience, 25(5), p.104220.; "Nano-domain behaviour in P3HT: PCBM nanoparticles, relating material properties to morphological changes." Solar energy materials and solar cells 117 (2013): 437-445.
Reviewer 2 Report
In the work entitled "Synthesis of organic semiconductor nanoparticles with different conformations by nanoprecipitation method," four stable conformations of organic semiconductors P3HT and PCBM were produced by nanoprecipitation, including blend and core-shell nanoparticles. Ecologically correct methods produced all nanoparticles in aqueous suspension without surfactants. DLS, TEM, SEM, UV-visible absorption spectroscopy, and UV-visible photoluminescence were used to study the nanoparticles' structural and optoelectronic properties (PL). This work offers the procedures for harnessing these colloids as a photoactive layer of organic photovoltaic devices that might interact with biological systems due to their features and surfactant-free aqueous suspensions. I would recommend it be published in Polymers after the following issues are addressed:
- The resolution of the figures (5 and 6) is not acceptable in the current form. Please check.
- The PCBM should be expressed as PC71BM in the whole manuscript.
- The authors should provide detailed information on P3HT, for the 93% should be Regio Regular >93% instead of purity. The molecule weight and PDI should be provided so the newcomers, whether in the same group, could repeat same results.
- Since the authors noted that the experimental parameters must be strictly controlled, I would like to ask whether the synthetic scale affects the data reproducibility. If so, the authors should revise the experimental procedure into a more specific version (including how much weight they use, polymer batch, etc.) so the result could be repeated.
- DMSO is UV-active. Did the authors measure the blank to ensure the data measurement?
- How many batches did the authors try? What is the deviation of DLS when using different batches of NPs? Can the authors provide the average DLS from different synthetic batches?
- I suggest the authors measure the film of four compositions using THF(spin coating on ITO). By comparing the UV curves of solid film with NPs, the differences could provide more information on the extent of the polymer/PC71BM aggregation.
- The sentence should be removed since this work has no PV application. "It is interesting that the absorption spectrum in the UV-Visible region is as broad as possible for photovoltaic applications since they will be able to absorb photons in a wider wavelength range." In addition, the authors should understand that both the range and absorption coefficient are essential for PV application, and the partial explanation could be misleading.
- Please revise Figure 5b. The normalized curves should show the highest point at 1 on the Y-axis. The authors seem to use wider range for the normalization calculation.
- For the introduction, several works of organic photovoltaic, core-shell synthesis, aggregation of polymer, and the analysis of PC71BM morphology should be cited—Nanotechnology 31 (2020) 375605 (doi.org/10.1088/1361-6528/ab9678); Organic Electronics 87 (2020) 105986(doi.org/10.1016/j.orgel.2020.105986)
Round 2
Reviewer 1 Report
The authors have addressed my concerns, and I recommend publication in the journal.